# Unraveling the Complexity: Exploring the Intersection of Panic Disorder, Dissociation, and Complex Post-Traumatic Stress Disorder

**DOI:** 10.3390/bs14030166

**Published:** 2024-02-22

**Authors:** Martina D’Angelo, Marta Valenza, Anna Maria Iazzolino, Grazia Longobardi, Valeria Di Stefano, Elena Lanzara, Giulia Visalli, Luca Steardo, Caterina Scuderi, Luca Steardo

**Affiliations:** 1Psychiatry Unit, Department of Health Sciences, University of Catanzaro Magna Graecia, 88100 Catanzaro, Italy; martina.dangelo001@studenti.unicz.it (M.D.); annamaria.iazzolino@studenti.unicz.it (A.M.I.); grazia.longobardi@studenti.unicz.it (G.L.); valeria.distefano@studenti.unicz.it (V.D.S.); giulia.visalli@studenti.unicz.it (G.V.); 2Department of Physiology and Pharmacology “Vittorio Erspamer”, Sapienza University of Rome, 00185 Rome, Italy; marta.valenza@uniroma1.it (M.V.); luca.steardo@uniroma1.it (L.S.); caterina.scuderi@uniroma1.it (C.S.); 3Department of Clinical Psychology, University Giustino Fortunato, 82100 Benevento, Italy; elenalanzara17@gmail.com

**Keywords:** panic disorder, complex posttraumatic disorder, dissociation, anxiety

## Abstract

Background: Patients with panic disorder (PD) may experience increased vulnerability to dissociative and anxious phenomena in the presence of repeated traumatic events, and these may be risk factors for the development of complex post-traumatic stress disorder (cPTSD). The present study aims to find out whether the presence of cPTSD exacerbates anxiety symptoms in patients suffering from panic disorder and whether this is specifically associated with the occurrence of dissociative symptoms. Methods: One-hundred-and-seventy-three patients diagnosed with PD were recruited and divided into two groups based on the presence (or absence) of cPTSD using the International Trauma Questionnaire (ITQ) scale. Dissociative and anxious symptoms were assessed using the Cambridge Depersonalization Scale (CDS) and Hamilton Anxiety Scale (HAM-A), respectively. Results: Significant differences in re-experienced PTSD (*p* < 0.001), PTSD avoidance (*p* < 0.001), PTSD hyperarousal (*p* < 0.001), and DSO dysregulation (*p* < 0.001) were found between the cPTSD-positive and cPTSD-negative groups. A statistically significant association between the presence of cPTSD and total scores on the HAM-A (*p* < 0.001) and CDS (*p* < 0.001) scales was found using regression analysis. Conclusions: This study highlights the potential link between dissociative symptoms and a more severe clinical course of anxiety-related conditions in patients with PD. Early intervention programs and prevention strategies are needed.

## 1. Introduction

### 1.1. Panic Attack Disorder: Understanding the Clinical Presentation

In the realm of contemporary psychiatry, significant attention has been devoted to anxiety disorders, particularly panic disorder (PD), which ranks among the most prevalent reasons for seeking specialized consultation within the context of mental disorders [1] as it is the most prevalent anxiety disorder in the general population of the Western world, boasting a yearly prevalence of 2–3% in Europe [1,2]. In the field of modern medicine, the historical progression of understanding PD has been marked by the challenge of distinguishing it from neurological, cardiological, and endocrinological syndromes. Even currently, the intricate clinical presentation, characterized by a combination of physical, cognitive, and behavioral symptoms, occasionally continues to pose challenges in terms of diagnostic evaluation [3,4]. Preclinical and clinical research provides substantial support for the concept that PD represents a fear response acquired through specific bodily sensations [5]. PD is a distressing psychiatric condition characterized by cognitive distortions, including the misinterpretation of benign physical or mental sensations. This disorder is closely associated with significant impairment in overall quality of life, encompassing marital, social, and occupational functioning [6] with effects akin to those observed in many other chronic psychiatric conditions [7,8,9].

A growing body of research suggests that PD and other anxiety disorders in adulthood can be viewed as manifestations of a vulnerability or predisposition to anxiety [10]. Although the median age of panic onset typically falls between 20 and 30 years, more than half of patients with PD report a history of anxiety, in various forms, dating back to childhood [2]. Moreover, PD often displays familiar patterns, and hereditary factors may confer susceptibility to anxiety [11]. Studies have reported increased rates of behavioral inhibition in children at risk of developing anxiety disorders [12,13]. Conversely, high rates of childhood and adult anxiety have been identified in parents of children characterized as having behavioral inhibition [14]. Such vulnerability to anxiety can also be influenced by environmental factors, including trauma, early learning, conditioning, and parental behavior [15]. Early observations of anxious and panicogenic phenomena underscore the close relationship between psychological experiences and associated somatic symptoms [16,17].

### 1.2. Impact of Trauma and Dissociation in Panic Attack Disorder

PD is associated with various psychiatric conditions, including depression and other anxiety disorders.

In the DSM-5, PD is the sole anxiety disorder that includes a dissociative symptom—derealization or depersonalization—as a criterion. Nevertheless, a diagnosis can be given even in the absence of depersonalization or derealization, which is often present in PD and can be considered a manifestation of dissociative experiences [18,19,20]. Acute dissociation may be associated with symptoms of PD [21]. Previous research on anxiety disorders has shown that both dissociative experiences and anxiety are often overlooked but contribute significantly to the overall illness burden [22].

Traumatic experiences, particularly childhood abuse, should be considered in the etiology of dissociation [23,24,25]. A recent review by Fung and colleagues demonstrated that a considerable number of patients with complex post-traumatic stress disorder (cPTSD) exhibited dissociative symptoms, with a clinically relevant dissociation rate of 76.9% [26]. Dissociation often functions as a defense mechanism to protect an individual’s psychological integrity against trauma [27,28,29,30,31,32]. There is a hypothesis that panic attacks can serve as traumatic stressors and comorbid PTSD aggravates the heightened vulnerability of individuals with PD to experiencing dissociative episodes because of traumatic stress [33]. To complicate this picture, both childhood trauma and dissociation are independently associated with other risk factors, such as suicide [34]. Dissociative symptoms can also play a decisive role in treatment resistance. This condition may not only hinder the progress of psychotherapy but may also impede a favorable response to pharmacotherapy [35]. Therefore, considering the relationship between PD dissociative symptomatology and trauma is relevant not only for diagnosis but also for choosing the appropriate therapeutic approach [36].

Studies have shown that complex trauma is associated with a higher severity of anxiety symptoms in some disorders, as high HAM-A scores have been observed in patients with an obsessive–compulsive disorder or bipolar disorder (BD) in the case of comorbid cPTSD. Moreover, in patients with BD, both studies showed a clear association between dissociative symptoms and cPTSD as well as with anxious characteristics [37,38,39]. A recently published non-patient survey study by Yang and colleagues showed that those who tested positive for cPTSD showed a higher incidence of anxiety and hypervigilance [40]. To our knowledge, no study has yet examined the impact of cPTSD on PD symptoms, particularly its contribution to the exacerbation of dissociative symptoms. Therefore, this study aimed to examine whether the presence of cPTSD comorbid with PD is associated with dissociative symptoms and whether it correlates with a greater severity of anxiety symptoms among individuals diagnosed with PD.

## 2. Materials and Methods

The design of our study is observational and cross-sectional, conducted within a naturalistic setting. We consecutively enrolled patients diagnosed with PD at the Department of Psychiatry at the “Magna Graecia” University of Catanzaro. Each participant received comprehensive information regarding the research objectives, personal data protection, and the assurance of privacy and anonymity. Participation in the study was voluntary. All patients provided written informed consent to participate after receiving a complete explanation of the study objectives and design. The study was conducted following the latest version of the Declaration of Helsinki and received approval from the Ethics Committee of the University of Catanzaro (approval number: 308/2020). Patients were recruited based on the following inclusion criteria:− Diagnosis of PD according to the Diagnostic and Statistical Manual of Mental Disorders (DSM-5) criteria, assessed through clinical interviews and the administration of psychometric scales (SCID-5-CV);− Age between 18 and 75 years.

The exclusion criteria, however, were:− Patient refusal to participate;− Presence of significant neurological and psychiatric disorders (e.g., epilepsy, intellectual disabilities, genetic syndromes with psychiatric manifestations);− Conditions that would hinder the completion of a comprehensive assessment, such as language barriers and severe specific cognitive impairments (e.g., dyslexia);− Lack of signed informed consent form.

Following recruitment, participants underwent clinical and psychopathological assessments. These evaluations were conducted during outpatient clinical visits, with the administration of rating scales either at the end of the visit or during separate appointments. The administration of psychometric rating scales and the collection of sociodemographic data were carried out by researchers, medical specialists in training, and PhD students. Each enrolled patient underwent a semi-structured clinical interview to gather clinical and anamnestic information. Sociodemographic and clinical data were collected using a specialized medical history questionnaire developed within our department. This questionnaire is administered by healthcare professionals and is divided into two sections: the first part focuses on collecting sociodemographic data, while the second part gathers information concerning the patient’s psychiatric history even the presence and nature of trauma [41]. After the initial screening phase, patients underwent the following psychometric scales to measure the presence of complex trauma, the severity of anxiety symptoms, and the presence of dissociative symptomatology. A brief description of these psychometric tools will be provided in the next paragraph.

### 2.1. Psychometric Tools

The rating scales employed in our study included the Cambridge Depersonalization Scale (CDS), the Hamilton Anxiety Scale (HAM-A), and the International Trauma Questionnaire (ITQ).

The CDS was developed to provide a standardized measure of depersonalization severity [42]. It comprises a self-administered questionnaire requiring participants to assess the frequency and intensity of depersonalization symptoms experienced over a specified period, typically in the past weeks or months [43]. The CDS enables the examination of correlations between depersonalization and factors such as traumatic experiences, genetic predisposition, stress, or other psychosocial events, thus contributing to the understanding of the underlying etiology [44]. The questionnaire consists of 29 items and is rated on two independent Likert scales: one for frequency (ranging from 0 to 4) and another for severity (ranging from 1 to 6). A total score is derived by summing all items’ scores. An intensity index for each item can also be calculated by adding the frequency and severity scores (ranging from 0 to 10) [45].

The HAM-A emerges as an invaluable evaluative tool for attaining an enhanced comprehension of anxiety. Comprised of 14 items, it encompasses a diverse array of anxiety manifestations, spanning from physiological to cognitive domains [46]. Its application facilitates a meticulous assessment of both somatic and cognitive dimensions of anxiety. Moreover, it demonstrates a high degree of sensitivity, rendering it particularly adept at discerning even subtle fluctuations in anxiety symptomatology over time [47]. The HAM-A is widely used and validated in clinical anxiety research settings, enabling the juxtaposition of findings derived from this metric with those of other investigations, thereby fostering a more nuanced elucidation of anxiety etiology and therapeutic interventions [47]. Each item is rated on a 5-point Likert scale, and the total score can range from 0 to 56, with a score of 18 or above indicating a pathological level of anxiety [48].

The ITQ is a self-administered 18-item questionnaire designed to explore interpersonal traumas and their impact on an individual’s development, distinguishing between post-traumatic stress disorder (PTSD) and complex PTSD (cPTSD) [49]. The ITQ represents a concise and straightforward tool designed to assess core aspects of PTSD and cPTSD, aligning with the organizational principles outlined in the International Classification of Diseases 11th Revision by the World Health Organization, providing both categorical diagnostic scores and dimensional severity scores [50]. A 5-point Likert scale, ranging from 0 (not at all) to 4 (very strongly), can be used to answer the questions. One of two symptoms from each of the three PTSD symptom clusters (re-experiencing, avoidance, and sense of current threat), as well as one of two symptoms from each of the three Disturbances in Self-Organization (DSO) clusters (affective dysregulation, negative self-concept, and relationship disturbances), must be confirmed to diagnose cPTSD. Therefore, the highest score for DSO and/or PTSD is 24 (range 0–24), whereas the maximum score for cPTSD is 48 (range 0–48). A value of ≥2 on the Likert scale is required for all items to be deemed present. A person can only receive one of the two diagnoses, namely PTSD or cPTSD [51].

The reliability coefficients of CDS, HAM-A, and ITQ are 0.785, 0.684, and 0.691, respectively.

### 2.2. Statistical Analysis

All data were stored in an electronic dataset. Descriptive statistical analyses were conducted to assess the distribution characteristics of sociodemographic and clinical variables within the sample. Continuous variables were presented as means and standard deviations (SD), while categorical variables were summarized as frequencies and percentages (%). Levene’s test was used to test for the homogeneity of variance. The differences between the two independent groups were analyzed using a two-tailed Student’s *t*-test.

The regression analysis was used to assess the association between cPTSD and the severity of anxious symptoms and dissociative experiences, considering as independent variables the HAM-A and CDS total scores, and as dependent variables the diagnosis of cPTSD. To evaluate potential multicollinearity issues, VIF and tolerance values were calculated indicating no significant multicollinearity concerns. No data were excluded from statistical analysis for any reason (e.g., outliers).

The Statistical Package for the Social Sciences version 26 (SPSS, Chicago, IL, USA) was used to perform all statistical analyses.

## 3. Results

Table 1 displays the clinical and socio-demographic characteristics of the sample. The final sample recruited for our study comprised 173 patients diagnosed with PD, out of which 83 (47.7%) were male, with an average age of 47.2 years (SD ± 14.0). Among the sample, 79 (45.4%) cohabited, and 136 (78.2%) had a college degree. Participants exhibited varying levels of PTSD symptoms, as assessed by the ITQ: those who experienced symptoms of trauma re-experiencing had a mean value of 1.56 (SD ± 1.55); individuals attempting to avoid trauma-related experiences scored an average of 1.71 (SD ± 1.55); and arousal states facilitating defensive reaction corresponded to an average of 1.77 (SD ± 1.52). Additionally, the mean score for all subjects with PTSD was 5.04 (SD ± 4.33). Participants exhibiting symptoms of Self-Organization Disorder (DSO) had the following scores: emotional dysregulation, 1.75 (SD ± 1.76); negative self-image, 1.41 (SD ± 1.47); and difficulties in relationship areas, 1.64 (SD ± 1.68). The overall DSO mean score was 4.80 (SD ± 4.75).

The CDS score describing depersonalization and derealization symptoms was 15.5 (SD ± 20.8). Further, the mean HAM-A score was 28.6 (SD ± 13.3).

Patients were interviewed using the ITQ, HAM-A, and CDS psychometric scales to determine whether or not they had cPTSD, the severity of their anxiety symptoms, and whether they suffered from dissociative symptoms. As shown in Table 2, one-hundred-and-thirty-five patients resulted positive for cPTSD from the ITQ, with mean values of PTSD re-experience of 1.94 (SD ± 1.55), PTSD avoidance of 2.09 (SD ± 1.54), and PTSD hyperarousal of 2.14 (SD ± 1.51). The mean scores of the three PTSD clusters and the three DSO clusters analyzed with the ITQ were, respectively, 6.17 (SD ± 4.26) and 6.08 (SD ± 4.66).

The statistical analysis using the *t*-test for independent samples shows that there are significant differences between PD patients with and without comorbid cPTSD. Differences between groups are statistically significant for all measured variables. Individuals with cPTSD experience more symptoms related to re-experiencing trauma, avoidance of trauma reminders, and heightened arousal, with an overall greater symptom severity (all *p* > 0.001 vs. cPTSD-negative group). Further, both the CDS and HAM-A scales yielded significantly higher total scores in individuals with cPTSD than non-cPTSD ones (*p* < *0*.001), highlighting the clinical relevance of this comorbidity. These findings suggest that cPTSD has a significant impact not only on typical PTSD symptoms but also on anxiety symptoms and depersonalization. This implies that individuals with PD and cPTSD experienced more severe anxiety symptoms and dissociative experiences compared to those without cPTSD (Table 3).

The result of the regression analysis carried out considering cPTSD data as the dependent variable and the total scores of HAM-A and the CDS data as independent variables shows a statistically significant association between the presence of cPTSD and total scores on both HAM-A and CDS (*p* < 0.001), as shown in Table 4. No significant multicollinearity issues were found in this regression analysis. Such a result indicates that both HAM-A and CDS scores are significant predictors of cPTSD.

## 4. Discussion

This study offers valuable insights into the intricate interplay of psychopathological dimensions in individuals affected by PD with comorbid cPTSD. Our research represents a pioneering endeavor in understanding the relationship between repeated traumatic stress over the lifetime of a person already suffering from panic disorder and the onset of dissociative symptoms in adulthood, alongside their correlation with increased anxiety symptomatology. Moreover, this study highlights the potential link between dissociative symptoms and a more severe clinical course of anxiety-related conditions, shedding light on the complex interplay between trauma and psychopathological outcomes. Our results revealed compelling findings that hold significant implications for both clinical practice and further research.

Given the importance of the long-term impact of traumatic experiences that may contribute to the manifestation of dissociative symptoms in people with PD, this is, to our knowledge, the first study to examine the relationship between traumatic experiences, PTSD, and the development of dissociative symptoms in a specific population of patients with PD.

Results presented in this study unveil a robust correlation between the dissociative dimension and the presence of trauma in individuals struggling with panic attacks. Notably, several clinical variables were identified as having a greater psychopathological burden in this population. These findings underscore the significance of recognizing and addressing dissociative symptoms in patients with PD, particularly those with a history of trauma. It is important to note that while dissociative disorders have a distinct diagnostic category, dissociative symptoms are common across various psychiatric conditions [25,52]. This study underlines that dissociative symptoms are not confined to a single diagnostic group but can influence the clinical course of various psychiatric disorders. Panic attacks, for instance, include dissociative symptoms as a criterion, emphasizing the need for comprehensive assessment and treatment strategies [53].

PD is a prevalent anxiety disorder and a significant risk factor for the development of broader psychopathology [1]. In clinical practice, patients with severe PD symptoms often present with comorbid dissociative disorders and a high burden of dissociative symptoms. These observations suggest that PD may be intertwined with a dissociative process, potentially indicative of complex dissociative disorders [54]. Interestingly, our study reveals a higher incidence of dissociative symptoms in patients with PD and cumulative traumatic exposure compared to individuals who have experienced only one trauma [23]. This heightened association between anxiety and dissociation may be attributed to the physical and physiological manifestations of PD symptoms, which, when accompanied by phenomena like derealization and depersonalization, can intensify the individual’s anxiety. A vicious cycle may ensue, perpetuated by the interpretation of these physical symptoms [55]. Panic attacks are frequently accompanied by dissociative symptoms, leading to brief periods of sensory deprivation or reduced sensory input. Moreover, individuals with PD often experience sleep disturbances, which can further contribute to the development of dissociative symptoms [56]. A more in-depth analysis of clinical variables has identified predisposing factors for the onset of dissociative symptoms. One study has shown that patients with PD and comorbid dissociative disorder, along with a high degree of dissociative symptoms, exhibit more severe panic symptoms [55,57]. However, no significant difference in dissociative experiences was found between PD patients and normal controls or individuals with other anxiety disorders [55].

Higher Dissociative Experiences Scale (DES) scores have been correlated with an increased frequency of lifetime affective decompensations, indicating a profound impact on individuals’ psychological well-being. Additionally, seasonality has been found to correlate with dissociative symptoms and a more challenging clinical trajectory [58]. Understanding and addressing these intricate dynamics requires the implementation of psychotherapeutic interventions tailored to structured trauma intervention [59]. Trauma, characterized by sudden and violent experiences, has the potential to profoundly affect an individual’s psychological functioning and adaptation [60]. Such experiences often trigger behaviors like avoidance or dissociation, which may become maladaptive if not effectively addressed. By delving deeper into these dynamics, clinicians can better comprehend the complexities of trauma-related symptomatology and its implications for therapeutic intervention. Dissociation, as measured by the DES, serves as a marker for the severity and complexity of trauma-related symptoms. Individuals with higher DES scores often experience a greater number of affective decompensations over their lifetime, highlighting the enduring impact of trauma on psychological health. Moreover, the influence of seasonality on dissociative symptoms underscores the nuanced nature of trauma-related psychopathology. Seasonal variations may exacerbate symptoms and contribute to a more challenging clinical course for individuals with trauma histories [61]. This underscores the importance of considering environmental factors in understanding and treating dissociative symptoms. To address these complex dynamics, psychotherapeutic interventions must prioritize structured trauma intervention. Trauma-focused therapies provide a framework for individuals to explore and process traumatic experiences in a safe and supportive environment [62]. By addressing the root causes of trauma-related symptoms, these interventions empower individuals to develop healthier coping strategies and achieve greater emotional well-being. Recognizing the potential for avoidance and dissociation as maladaptive coping mechanisms is essential in therapeutic practice. Left unaddressed, these behaviors can perpetuate cycles of distress and hinder individuals’ ability to engage fully in their lives [37,38,63]. Through targeted interventions, clinicians can help individuals cultivate resilience and develop adaptive strategies for managing trauma-related symptoms [64]. The intersection of dissociative symptoms, affective decompensations, and seasonality underscores the multifaceted nature of trauma-related psychopathology [65]. By integrating structured trauma intervention into psychotherapeutic practice, clinicians can effectively address the complex dynamics underlying these symptoms and support individuals on their journey toward healing and recovery [66].

Combining trauma-focused psychotherapy with cognitive-behavioral approaches is recommended [67]. Trauma-focused cognitive behavioral therapy (CBT) plays a crucial role in helping patients identify and modify distorted thinking patterns related to traumatic events, self-perception, and the world. It equips individuals with strategies to manage anxiety and negative emotions effectively, ultimately reducing persistent hyperarousal symptoms [62]. Exposure therapy, a key component of trauma-focused CBT, is based on principles of habituation and information processing. This assumption holds for PTSD or childhood trauma, but some studies, particularly those conducted on adolescent populations, demonstrate that such interventions are less effective when cPTSD is comorbid [68]. This should prompt research in psychotherapeutic fields to focus not only on singular traumas but also on what can be defined as cumulative trauma, as seen in cases of cPTSD [69]. Prolonged exposure, either in vivo or through imaginative exercises, allows individuals to reprocess traumatic information, modify their responses, and better manage anxiety [70]. Instead, this study’s contribution lies in discriminating symptomatology that has been little investigated, such as cPTSD, and guiding clinicians toward personalized therapy. To elaborate, in addition to the standard prescription of serotonin reuptake inhibitors or CBT, which are highly effective in treating PD alone, when it co-occurs with cPTSD, treatments should be augmented with interventions focused on trauma, both psychotherapeutic and psychosocial (e.g., EMDR, dialectical behavior therapy, etc.) [71]. Furthermore, the presence of cPTSD, defined as exposure to stressful events from which the patient cannot escape, may exacerbate dissociative symptoms characterized by the re-experiencing of traumatic memories, necessitating targeted therapeutic attention. This unmet need is supported by biological and clinical evidence. Firstly, a considerable bilateral amygdala–hippocampal dysfunction was detected, rendering certain types of psychotherapies ineffective. This is believed to be due to dissociative symptoms resulting from exposure to traumatic events [72].

The findings of the regression analysis highlight the importance of recognizing and addressing cPTSD in individuals with PD, as it significantly contributes to the severity of anxious and dissociative symptoms. By identifying these associations, clinicians can better tailor interventions to address the complex symptomatology observed in this population. Moreover, addressing multicollinearity concerns enhances the robustness of the regression analysis results, ensuring that the associations observed are not confounded by intercorrelations between independent variables. Therefore, addressing issues of multicollinearity and clearly defining independent and dependent variables in regression analysis strengthens the validity and interpretability of the study findings.

It is, however, necessary to recognize the limitations of our study. The relatively small sample size may impact the generalizability of our findings. The cross-sectional design and the relatively small sample size may have limited the generalizability of the findings. Additionally, the potential for recall bias due to the completeness and precision of patient accounts and disease states cannot be overlooked. The study mainly relied on the analysis of lifetime variables that were not affected by clinical severity at the moment of assessment. Future prospective longitudinal studies are expected to clarify the association between cPTSD and PD. Further, another important issue is analyzing the different patterns of affective dysregulation that might emerge from cPTSD. Furthermore, another important limitation is the lack of outlier analysis.

## 5. Conclusions

In conclusion, our study demonstrates that PD and cPTSD frequently co-occur, resulting in a complex clinical picture characterized by diverse symptomatology. The findings highlight the importance of recognizing and addressing cPTSD as an integral component of diagnostic and therapeutic considerations in individuals with PD. Tailored interventions targeting the unique symptom profiles associated with cPTSD are imperative for improving clinical outcomes. Future research should delve deeper into the underlying mechanisms of this comorbidity and explore innovative treatment modalities to enhance the well-being of affected individuals. Endeavors should explore the dissociative dimension in comorbidity with post-traumatic stress disorder (PTSD) and delve into the underlying mechanisms of this correlation. Moreover, investigating the most effective therapeutic strategies to mitigate the impact of repetitive traumatic events on disease progression represents a crucial avenue for further investigation. By shedding light on the complex interplay between complex trauma, PD, and dissociative symptoms, this study paves the way for enhanced clinical assessment and intervention, ultimately aiming to improve the overall well-being and outcomes of individuals affected by these conditions, structuring psychosocial and psychotherapeutic interventions that can reduce symptom burden and personalize treatment.

## Figures and Tables

**Table 1 behavsci-14-00166-t001:** Socio-demographic characteristics of the sample (*n* = 173 patients).

Male sex yes (*n* %)	83	47.7%
Cohabitant yes (*n* %)	79	45.4%
Diploma yes (*n* %)	136	78.2%
Age m (ds)	47.2	±14.0
Age of onset m (ds)	27.5	±11.1
Age of first contact m (ds)	30.3	±11.0
PTSD relived m (ds)	1.56	±1.55
PTSD avoidance m (ds)	1.71	±1.55
PTSD hyperarousal m (ds)	1.77	±1.52
PTSD total score m (ds)	5.04	±4.33
DSO dysregulation m (ds)	1.75	±1.76
DSO negative self-image m (ds)	1.41	±1.47
DSO relational disorders m (ds)	1.64	±1.68
DSO total score m (ds)	4.80	±4.75
CDS total score m (ds)	15.5	±20.8
HAM total score m (ds)	28.6	±13.3

CDS: Cambridge Depersonalization Scale; DSO: Self-Organization Disorder (subitem International Trauma Questionnaire), HAM-A: Hamilton Anxiety Rating Scale; PTSD: posttraumatic stress disorder (sub-item of International Trauma Questionnaire); *n*: total number; SD: standard deviation; %: percentage.

**Table 2 behavsci-14-00166-t002:** Descriptive analysis of the results of the psychometric scales in the sample (*n* = 173), divided based on the presence or absence of cPTSD.

	cPTSD	*n*	Mean	SD	SE
PTSD re-experience	0	39	0.26	0.44	0.07
	1	135	1.94	1.55	0.13
PTSD avoidance	0	39	0.36	0.49	0.08
	1	135	2.10	1.54	0.13
PTSD hyperarousal	0	39	0.49	0.56	0.09
	1	135	2.14	1.51	0.13
PTSD total score	0	39	1.10	0.88	0.14
	1	135	6.18	4.26	0.37
DSO dysregulation	0	39	0.28	0.46	0.07
	1	135	2.18	1.77	0.15
DSO total score	0	39	0.39	0.59	0.09
	1	135	6.08	4.66	0.40
CDS total score	0	39	3.46	1.68	0.27
	1	135	18.9	22.4	1.93
HAM-A total score	0	39	11.23	2.66	0.43
	1	135	33.59	10.65	0.91

CDS: Cambridge Depersonalization Scale; DSO: Self-Organization Disorder (subitem International Trauma Questionnaire), HAM-A: Hamilton Anxiety Rating Scale; PTSD: posttraumatic stress disorder (subitem International Trauma Questionnaire); 0 = no; 1 = yes.

**Table 3 behavsci-14-00166-t003:** Statistical analysis of the results of the psychometric scales in the sample (*n* = 173), between groups with or without cPTSD.

	*t*	df	*p*
PTSD re-experience	6.68	172	<0.001
PTSD avoidance	6.93	172	<0.001
PTSD hyperarousal	6.69	172	<0.001
PTSD total score	7.37	172	<0.001
DSO dysregulation	6.61	172	<0.001
DSO total score	7.60	172	<0.001
CDS total score	4.3	172	<0.001
HAM-A total score	12.97	172	<0.001

CDS: Cambridge Depersonalization Scale; DSO: Self-Organization Disorder (subitem International Trauma Questionnaire); HAM-A: Hamilton Anxiety Rating Scale; PTSD: posttraumatic stress disorder (subitem International Trauma Questionnaire).

**Table 4 behavsci-14-00166-t004:** Regression analysis with cPTSD as the dependent variable and the total scores of HAM-A and CDS as independent variables, considering the collinearity statistics.

Dependent Variable cPTSD	*t*	*p*	VIF	Tolerance
CDS total score	5.62	< 0.001	1.42	0.703
HAM-A total score	19.49	<0.001	1.42	0.703

CDS: Cambridge Depersonalization Scale; HAM-A: Hamilton Anxiety Rating Scale; cPTSD: complex post-traumatic stress disorder. *p*-values indicate statistical significance.

## Data Availability

The data that support the findings of this study are available from the corresponding author, upon reasonable request.

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
