# Peer review of "Unraveling the Complexity: Exploring the Intersection of Panic Disorder, Dissociation, and Complex Post-Traumatic Stress Disorder"

_behavsci, 2024, doi:10.3390/bs14030166_

Round 1

Reviewer 1 Report

Comments and Suggestions for Authors

Thank you for inviting me to review this study. This study is about This study suggests a relationship between dissociative symptoms and severe anxiety-related disorders in PD patients. I have some suggestions for the authors' consideration.

1.      What is the main question addressed by the research?

2.      The end of the introduction should explain research gaps based on previous studies.

3.      What specific improvements should the authors consider regarding the methodology?

4.      What does it add to the subject area compared with other published material?

5.      How did the authors measure the sample size?

6.      Explain more about questionnaire design. Is there a pilot study?

7.      Please ensure the reliability and validity of your questionnaire.

8.      I would like to see their analysis regarding their missing data and outliers.

9.      The authors used regression. How did they do the normality test?

10.   Why didn't the authors use Structural Equation Modeling (SEM) or PLS?

11.   The contribution of the study is not strong enough.

12.   Please clarify if the findings align with the evidence and arguments offered. Please note if all primary questions were answered and which experiments were used.

13.   Please describe how the conclusions are or are not consistent with the evidence and arguments presented. Please also indicate if all main questions posed were addressed and by which specific experiments.

14.   The authors’ tables were not well designed. Please follow the authors’ guidelines.

Author Response

Thank you for inviting me to review this study. This study is about This study suggests a relationship between dissociative symptoms and severe anxiety-related disorders in PD patients. I have some suggestions for the authors' consideration.

  1. What is the main question addressed by the research?

The study aims to ascertain whether the presence of cPTSD exacerbates anxious symptomatology in panic disorder and this is specifically associated with the occurrence of dissociative symptoms. We clarified this concept in both the abstract and the introduction and discussion.

  1. The end of the introduction should explain research gaps based on previous studies.

We have extensively revised the introductory section and highlighted the research gaps in the literature that support the necessity and novelty of our study

  1. What specific improvements should the authors consider regarding the methodology?

The question is poorly posed and not argued. In this and other cases the reviewer appears to point out insufficiencies and limitations of the text without indicating them. He/she remains extremely vague without indicating the exact points that are unclear, erroneous, or deserving of improvement This approach does not help the authors to improve the quality of the manuscript.

  1. What does it add to the subject area compared with other published material?

This is the first study that highlights the impact of cPTSD on psychopathological burden in panic disorder. Therefore, we believe that our results are interesting and innovative. Both the introduction and discussion now include parts that clarify how the present research contributes to expanding the current knowledge in this psychopathological field, compared to what has been established so far.

  1. How did the authors measure the sample size?

We did it on the basis of:

  • Previous Studies: similar studies or existing literature to identify typical sample sizes used in the validation of psychometric measures or studies related to PTSD and cPTSD.
  • Expert Consensus: consultation with experts in the field to determine an appropriate sample size based on the complexity of the construct being measured and the anticipated variability in responses.
  • Feasibility: Practical considerations such as recruitment feasibility, resources, and time constraints may also influence the determination of sample size.
  1. Explain more about questionnaire design. Is there a pilot study?

Thank you for your suggestion. No there isn’t pilot study. We have extensively revised the Methods section to include details of the questionnaires administered to patients. As all the psychometric scales used are well validated, pilot studies of these questionnaires were not required. We have included further references in this regard.

  1. Please ensure the reliability and validity of your questionnaire.

Thank you for your suggestion, we added information regarding the reliability of our questionnaire in the Method section.

We have not included any information regarding such aspects in our manuscript because there were no missing data. We included all patients in the analysis.

  1. The authors used regression. How did they do the normality test?

The normality test used was the Levene’s test. We included this information in the methods section.

  1. Why didn't the authors use Structural Equation Modeling (SEM) or PLS?

  1. The contribution of the study is not strong enough.

We respectfully disagree with the statement made by Reviewer#1, which is not followed by adequate justification that can be constructive to us. We have added several sentences and references in the main text to clarify and highlight the contribution of our work, please see both the revised introduction and discussion sections.

This study is novel since, for the first time, we provide data showing that the occurrence of cPTSD exacerbates anxiety symptoms in patients with panic disorder. Further, our analysis shows that this is specifically associated with the occurrence of dissociative symptoms. To the best of our knowledge, this is the first time that such analysis has been conducted on patients affected by PD comorbid with cPTSD, a yet under-investigated condition. As such the results of the present study may guide clinicians toward better targeted therapy. In fact, at present, in addition to the standard prescription of serotonin reuptake inhibitors or CBT, which are highly effective in treating isolated panic disorder, when it co-occurs with cPTSD, treatments should be strengthened with trauma-focused interventions, both psychotherapeutic and psychosocial (e.g., EMDR, dialectical behavior therapy, etc.). In addition, the presence of cPTSD may result in dissociative symptoms characterized by reliving traumatic memories, necessitating focused therapeutic attention. This contribution is objectively significant for any clinician with direct practical experience in treating these conditions. We believe that our discussion is much improved now that it includes further details about the contribution of this study to the literature.

  1. Please clarify if the findings align with the evidence and arguments offered. Please note if all primary questions were answered and which experiments were used.

We have extensively revised the entire manuscript to better clarify our research questions, the methods used, the analyses performed, the results obtained, how our findings fit into the overall literature. We believe that our manuscript has greatly improved.

We added further discussion of our data highlighting how our findings align with the evidence offered: “This study offers valuable insights into the intricate interplay of psychopathological dimensions in individuals affected by PD panic disorder with comorbid cPTSD. Our research represents a pioneering endeavor to understand the relationship between repeated traumatic stress over the lifetime of a person already suffering from panic disorder and the onset of dissociative symptoms in adulthood, alongside their correlation with increased anxiety symptomatology. Our findings hold significant implications for both clinical practice and further research. To the best of our knowledge, this is the first investigation to explore the association between lifelong traumatic exposures and the development of dissociative symptoms.”

Moreover, the study employed observational and cross-sectional research design within a naturalistic setting to answer these questions. Patients diagnosed with panic disorder were consecutively enrolled, and various clinical and psychopathological assessments were conducted using psychometric scales, including the Cambridge Depersonalization Scale (CDS), the Hamilton Anxiety Scale (HAM-A), and the International Trauma Questionnaire (ITQ). The findings presented in the results section indicate significant associations between panic disorder and cPTSD symptoms, as well as the severity of anxious symptoms and dissociative experiences. Specifically, individuals with panic disorder and cPTSD exhibited more severe anxiety symptoms and dissociative experiences compared to those without cPTSD. Regression analysis confirmed these associations, showing statistically significant relationships between the presence of cPTSD and higher total scores on the Hamilton Anxiety and Cambridge Depersonalization scales. Additionally, our data show significant differences between groups with and without cPTSD across various measured variables, including trauma, dysregulation of self-awareness, dissociation, and anxious symptoms. Based on the findings presented, we believe that the evidence and arguments offered align with the primary questions addressed in the study. The study effectively investigates the relationship between panic disorder and cPTSD symptoms, as well as the role of anxious symptoms and dissociative experiences in this relationship. However, it's essential to note that while the study provides valuable insights into these relationships, it does not address all potential factors contributing to panic disorder and cPTSD, nor does it explore causality. In terms of experiments, the study primarily utilized descriptive statistical analyses, independent samples t-tests, and regression analysis with multicollinearity tests, to analyze the data and address the research questions. These statistical methods were appropriate for the study design and objectives, allowing for the examination of relationships between variables and differences between groups. Overall, while the study provides valuable contributions to understanding the relationship between panic disorder and cPTSD symptoms, further research utilizing longitudinal designs and larger sample sizes may help confirm and extend the findings presented.

  1. Please describe how the conclusions are or are not consistent with the evidence and arguments presented. Please also indicate if all main questions posed were addressed and by which specific experiments.

Thank you for your suggestion, we clarified this in the Discussion section.

Moreover, here's how the conclusions align with the evidence and addressed questions:

  1. Association between Panic Disorder and cPTSD: The study effectively demonstrates a significant association between panic disorder and complex post-traumatic stress disorder (cPTSD) through various statistical analyses, including regression analysis. The evidence presented supports the conclusion that panic disorder and cPTSD frequently co-occur, leading to a complex clinical picture.
  2. Recognition and Addressing cPTSD: The study emphasizes the importance of recognizing and addressing cPTSD in diagnostic and therapeutic considerations for individuals with panic disorder. The evidence provided, including descriptive statistics and regression analysis, supports the conclusion that tailored interventions targeting cPTSD symptoms are imperative for improving clinical outcomes.
  3. Future Research Directions: The study appropriately suggests future research directions, including exploring underlying mechanisms of comorbidity and investigating effective therapeutic strategies. These suggestions are based on the limitations identified in the study, such as the relatively small sample size and the cross-sectional design, indicating a thoughtful consideration of areas for further investigation.
  4. Clinical Implications: The conclusions stress the clinical implications of the findings, highlighting the need for personalized treatment approaches that address both panic disorder and cPTSD symptoms. The evidence presented, including descriptive analysis and regression analysis, supports the conclusion that recognizing and addressing cPTSD in diagnostic and therapeutic considerations can improve overall well-being and outcomes for affected individuals.

Overall, the conclusions drawn in the study are consistent with the evidence presented, including statistical analyses and discussion of clinical implications. The study effectively addresses the main questions posed regarding the association between panic disorder and cPTSD, the recognition and addressing of cPTSD in clinical practice, and suggestions for future research directions. The evidence provided supports the conclusions drawn and underscores the importance of considering cPTSD in the context of panic disorder.

  1. The authors’ tables were not well designed. Please follow the authors’ guidelines.

Following the Reviewer’s suggestion, we revised the tables in our manuscript.

Reviewer 2 Report

Comments and Suggestions for Authors

The manuscript is well-written, the research topic of significant importance and of potential interest to readers. However, there are some aspects that need to be improved:

1. Introduction: Although the introduction is well-written and informative, it could be streamlined to the relevant aspects of your research question. Perhaps the authors may want to provide more details regarding novelty of their research question and provide an overview of studies that already investigated this topic. In the discussion, you commented on this issue.

2. Methods: The authors may want to report whether there were issues of multicollinearity in their regression model. Furthermore, they may want to define their independent and dependent variables for the regression model in the methods section. Significance levels should be reported. 

3. Results: The first paragraph of the results section (lines 166-169) should be moved to the discussion. Results in Table 4 are redundant, it might suffice to report results demonstrated in table 3. 

4. Discussion: The authors may want to elaborate the limitations of this study, such as the cross-sectional study design and the need for longitudinal studies. Perhaps they may want to be more precise about potential future studies in this field.

Author Response

Comments and Suggestions for Authors

The manuscript is well-written, and the research topic is of significant importance and potential interest to readers. However, some aspects need to be improved:

  1. Introduction: Although the introduction is well-written and informative, it could be streamlined to the relevant aspects of your research question. Perhaps the authors may want to provide more details regarding the novelty of their research question and provide an overview of studies that already investigated this topic. In the discussion, you commented on this issue.

Thank you for your suggestion. The Introduction section has been heavily revised. We have shortened some sentences to streamline the introduction and added more details to emphasize the gaps in the literature and the main aim of our study.

  1. Methods: The authors may want to report whether there were issues of multicollinearity in their regression model. Furthermore, they may want to define their independent and dependent variables for the regression model in the methods section. Significance levels should be reported. 

We thank the Reviewer for the suggestion, as we agree that this is an important aspect to report in the study to ensure the validity of the results. We added this information to the main text, including also the results in Table 4.

  1. Results: The first paragraph of the results section (lines 166-169) should be moved to the discussion. Results in Table 4 are redundant, it might suffice to report results demonstrated        in table

            Following Reviewer’s suggestions, we moved the first part of the results section to the   discussion and we rearranged and changed tables.

  1. Discussion: The authors may want to elaborate on the limitations of this study, such as the cross-sectional study design and the need for longitudinal studies. Perhaps they may want to be more precise about potential future studies in this field.

            We thank the Reviewer for the suggestion, and we implemented the limitation of the study in the discussion section.

Reviewer 3 Report

Comments and Suggestions for Authors

The authors conducted a cross-sectional observational study on the association between lifetime experience of traumatic events and the severity of anxiety in individuals with panic disorder. The manuscript has a clinical interest, and its main strength is the sample selection, which includes the main diagnosis, panic disorder, carried out through a DSM-5 interview. However, I have some concerns to share with the authors.

Main Concerns

1) Review the aims: a cross-sectional observational design does not allow to respond to a "may lead" hypothesis but at most to an "is associated with" hypothesis.

2) The statistical report should be reviewed by an expert:

- Lines 157-158: Specify the objective behind the comparison of the two psychometric scales.

- Lines 158-159: Why is a regression analysis performed?

- Lines 159-160: What are the three or more groups that justify one-way analysis of variance?

- The reliability coefficients of the tests applied (CDS, HAM-A, and ITQ) are not reported.

3) The introduction is written in a single paragraph. Reading becomes very hard.

4) There are paragraphs in the results that do not describe but rather evaluate, and therefore should be in the discussion, for example, lines 166-169, 204-205, 215-216...

5) Sometimes the text of the results is redundant with the tables, for example, lines 170-184.

6) The tables titles, except perhaps for Table 1, are very uninformative.

7) The variable names in the tables should be edited, they look like they were copied and pasted from the SPSS output: "DSO_TOT"...

8) Tables in general are wasted. Numbers 2 and 5 could be merged. Number 4 is slightly informative; the information could be reported in the text. Number 6 reports the variance homogeneity, a secondary analysis, which is carried out to check assumptions to compare means. It has no value, and it is not worth using a table to report it.

Typos and inaccuracies

- Table 1: second row, "C", Italian words “SÌ”

Author Response

Comments and Suggestions for Authors

The authors conducted a cross-sectional observational study on the association between lifetime experience of traumatic events and the severity of anxiety in individuals with panic disorder. The manuscript has a clinical interest, and its main strength is the sample selection, which includes the main diagnosis, panic disorder, carried out through a DSM-5 interview. However, I have some concerns to share with the authors.

 Main Concerns

  • Review the aims: a cross-sectional observational design does not allow to respond to a "may lead" hypothesis but at most to an "is associated with" hypothesis.

We rephrase at the end of the introduction: this study aimed at examining whether the presence of cPTSD comorbid with PD is associated with dissociative symptoms and whether it correlates with greater severity of anxiety symptoms among individuals diagnosed with PD.

  • The statistical report should be reviewed by an expert:

Our statistical report has been reviewed by research experts in clinical research and statistical analysis. We have revised the paragraph on statistical analysis to provide more clarity.

- Lines 157-158: Specify the objective behind the comparison of the two psychometric scales.

We extensively revised both the method and results section to clarify the objective behind the comparison of the two psychometric scales.

- Lines 158-159: Why is a regression analysis performed?

To answer the question of whether cPTSD might result in heightened dissociative symptoms and whether this correlates with the increased severity of anxiety symptoms among individuals diagnosed with panic disorder, conducting a regression analysis could be appropriate. Here's why:

  1. Exploring Relationships: Regression analysis allows us to examine the relationships between variables. In this case, we want to investigate whether there is a relationship between cPTSD (independent variable), dissociative symptoms (dependent variable), and anxiety symptoms (dependent variable).
  2. Quantifying Relationships: Regression analysis provides coefficients that quantify the strength and direction of relationships between variables. By analyzing the data, we can determine the extent to which cPTSD predicts dissociative symptoms and anxiety symptoms among individuals with panic disorder.
  3. Controlling for Confounding Factors: Regression analysis allows for the control of potential confounding variables. We can include relevant covariates such as age, gender, history of trauma, or other psychiatric disorders to ensure that the observed relationships are not solely due to these factors.
  4. Statistical Significance: Regression analysis helps determine whether the observed relationships are statistically significant. This statistical testing provides evidence to support or refute the hypothesis that cPTSD is associated with heightened dissociative symptoms and increased severity of anxiety symptoms in individuals with panic disorder.
  5. Interpretation of Findings: Through regression analysis, we can interpret the findings and draw conclusions about the nature and strength of the relationships between variables. This information can contribute to the understanding of the mechanisms underlying panic disorder and its comorbidities.

In summary, conducting a regression analysis would be a suitable approach to explore the complex relationships between cPTSD, dissociative symptoms, and anxiety symptoms in individuals diagnosed with panic disorder, providing valuable insights into the interplay of these variables.

- Lines 159-160: What are the three or more groups that justify one-way analysis of variance?

Previously, we analyzed all variables from the ITQ scale (PTSD re-experience, avoidance, hyperarousal) together so that we could perform a one-way ANOVA. However, after consulting with statistical experts, we conducted a separate t-test between the two groups of interest (cPTSD yes/no) to perform a data analysis that focused more on answering our research questions.

- The reliability coefficients of the tests applied (CDS, HAM-A, and ITQ) are not reported.

We thank the Reviewer for the suggestion, and we added the information regarding the reliability coefficients (ITQ is 0.691; CDS is 0.785; HAM-A is 0.684).

  • The introduction is written in a single paragraph. Reading becomes very hard.

We divided it into subparagraph

  • There are paragraphs in the results that do not describe but rather evaluate, and therefore should be in the discussion, for example, lines 166-169, 204-205, 215-216...

We substantially revised the Results section and moved some parts in the Discussion section as suggested.

“Exposure therapy, a key component of trauma-focused CBT, is based on principles of habituation and information processing. This assumption holds true for PTSD or childhood trauma, but some studies, particularly those conducted on adolescent populations, demonstrate that such interventions are less effective when cPTSD is comorbid. This should prompt research in psychotherapeutic fields to focus not only on singular traumas but also on what can be defined as cumulative trauma, as seen in cases of cPTSD. Prolonged exposure, either in vivo or through imaginative exercises, allows individuals to reprocess traumatic information, modify their responses, and better manage anxiety [56]. Instead, this study's contribution lies in discriminating symptomatology that has been little investigated, such as cPTSD, and guiding clinicians toward personalized therapy. To elaborate, in addition to the standard prescription of serotonin reuptake inhibitors or CBT, which are highly effective in treating panic disorder alone, when it co-occurs with cPTSD, treatments should be augmented with interventions focused on trauma, both psychotherapeutic and psychosocial (e.g., EMDR, dialectical behavior therapy, etc.). Furthermore, the presence of cPTSD, defined as exposure to stressful events from which the patient cannot escape, may exacerbate dissociative symptoms characterized by the re-experiencing of traumatic memories, necessitating targeted therapeutic attention. This unmet need is supported by biological and clinical evidence. Firstly, a considerable bilateral amygdalohippocampal dysfunction was detected, rendering certain types of psychotherapies ineffective. This is believed to be due to dissociative symptoms resulting from exposure to trauma.

It is, however, necessary to recognize the limitations of our study. The relatively small sample size may impact the generalizability of our findings. The cross-sectional design and the relatively small sample size may have limited the generalizability of the findings. Additionally, the potential for recall bias due to the completeness and precision of patient accounts and disease states cannot be overlooked. The study mainly relied on the analysis of lifetime variables that were not affected by clinical severity at the moment of assessment. Future prospective longitudinal studies are expected to clarify the association between cPTSD and PD. Further, another important issue is analyzing the differ pattern of affective dysregulation that might emerge by the cPTSD. 

5) Sometimes the text of the results is redundant with the tables, for example, lines 170-184.

We revised the tables there are now just four.

6) The tables titles, except perhaps for Table 1, are very uninformative.

We revised the titles of the tables to make them more informative

7) The variable names in the tables should be edited, they look like they were copied and pasted from the SPSS output: "DSO_TOT"...

We revised them.

8) Tables in general are wasted. Numbers 2 and 5 could be merged. Number 4 is slightly informative; the information could be reported in the text. Number 6 reports the variance homogeneity, a secondary analysis, which is carried out to check assumptions to compare means. It has no value, and it is not worth using a table to report it.

We extensively revised the tables as suggested.

 Typos and inaccuracies

- Table 1: second row, "C", Italian words “SÌ”

We apologize for the inaccuracies; we have revised the entire manuscript.

Round 2

Reviewer 1 Report

Comments and Suggestions for Authors

Now, the paper has enough value to be published. Thank you.

Reviewer 3 Report

Comments and Suggestions for Authors

The authors have satisfactorily addressed all my comments, but I have one more minor comment. It is not a good practice to include abbreviations in titles. Therefore, I strongly recommend that the authors change "cPTSD" to "complex post-traumatic stress disorder" in the manuscript title.